# The Effect of Value Innovation in the Superior Performance and Sustainable Growth of Telecommunications Sector: Mediation Effect of Customer Satisfaction and Loyalty

Mohammed A. Hajar [1], Ammar Ahmed Alkahtani [1,*], Daing Nasir Ibrahim [2], Mohammed A. Al-Sharafi [3,*], Gamal Alkawsi [1], Noorminshah A. Iahad [3,4], Mohd Ridzuan Darun [2] and Sieh Kiong Tiong [1]

[1] Institute of Sustainable Energy (ISE), Universiti Tenaga Nasional, Kajang 43000, Malaysia; eng.mohammed.hajar@gmail.com (M.A.H.); gamal.abdulnaser@uniten.edu.my (G.A.); siehkiong@uniten.edu.my (S.K.T.)
[2] Faculty of Industrial Management, Universiti Malaysia Pahang, Lebuhraya Tun Razak, Kuantan 26300, Malaysia; daing@ump.edu.my (D.N.I.); mridzuand@ump.edu.my (M.R.D.)
[3] Department of Information Systems, Azman Hashim International Business School, Universiti Teknologi Malaysia, Skudai 81310, Malaysia; minshah@utm.my
[4] Information Systems, Faculty of Science and Technology, Universitas Airlangga, Surabaya 60115, Indonesia
[*] Correspondence: ammar@uniten.edu.my (A.A.A.); alsharafi@ieee.org (M.A.A.-S.)

**Abstract:** The telecommunications industry has recognized innovation as the key to growth and survival. Globalization, liberalization, and privatization, the terms most commonly associated with this sector, have resulted in fierce competition, making it more difficult for telecommunications firms to increase their market share and, as a result, their customer base, sales volume, and, ultimately, profits. The new success strategy is differentiation through innovation, with the aim of breaking out of the competition and creating an uncontested market. This can be accomplished by providing high-value, innovative services that result in customer satisfaction and promote customer loyalty. The primary goal of this study was to create and validate a conceptual model of value innovation and its impact on firm performance and long-term growth by examining the mediation effect of customer satisfaction and loyalty. The empirical analysis results were based on 304 respondents who completed a paper-based survey provided to employees of Yemeni mobile service providers using a convenience non-probability sampling technique. SmartPLS 3 was used to test the hypothesized relationships using partial least squares structural equation modeling (PLS-PM). As a result, the findings empirically validated the theoretical research model, confirming the importance of the value innovation approach to achieving company performance and long-term growth by promoting customer satisfaction and loyalty. Finally, we have provided a discussion of the study's theoretical contributions, managerial implications, and future research directions.

**Keywords:** value innovation; superior performance; sustainable growth; telecommunications sector; customer satisfaction; customer loyalty

## 1. Introduction

In today's highly competitive business environment, maintaining a firm's success and survival has become a challenging task. Organizations are striving to satisfy the changing needs and rising expectations of customers. Accelerating innovation is increasingly important to the success and survival of businesses [1,2]. However, innovation per se has diverged into a wide range of domains and perspectives, making it more complicated for firms to identify their suitable paths to sustainable growth and survival. In general, innovation can take the form of a new product, service, technology, method of production, market, or management system [3,4]. So far, the key to a firm's success has been its strategic thinking and management capabilities in extracting value from new business opportunities and, as a result, making a significant change in the market, technology, and operation [5].

Organizations must promptly deliver the appropriate number of services and products to clients in order to effectively meet their changing needs [3]. Innovation is vital not only for a company's survival and growth but also for the nation and region's economic growth and progress [6]. According to Schumpeter [7] and Schumpeter [8], "innovation leads to the development and growth of the economy, and eventually to prosperity and wealth". As a result, with certain substantial sectoral and regional advancements, innovation is becoming a primary priority [9].

According to this viewpoint, organizations are becoming more innovative in their competition strategies, increasing the need for breakthrough innovation that imposes business differentiation, provides unprecedented value to customers, and creates intangible resources (competitive advantage) in order to achieve long-term superior performance and sustainable growth. According to [10], "the logic of value innovation focuses on creating new uncontested market space for both customers and firms by enabling business differentiation, making the competition irrelevant, and creating new uncontested market space" [11]. Value innovation assists firms in breaking out of the value-cost tradeoff of fierce competition by focusing on making the competition irrelevant through providing a quantum leap in value, rather than scattering their resources and capabilities in attempts to beat the existing competition [12]. As a disruptive innovation, value innovation may occur with or without technological breakthroughs as it aims to effectively utilize the technological and managerial opportunities to link innovation to value, create new demands, and change the market to render the competition irrelevant [11,13]. In this context, this study considers value innovation as a strategy that embraces the activities of the entire system of a company to achieve a quantum leap in value for buyers, as well as profitable growth and a competitive advantage for companies [14].

Previous studies have focused strongly on innovation adoption [15,16], technological innovation, and the innovation climate [17,18]. However, the literature has scarcely investigated the linking of value innovation with customer satisfaction and loyalty as an approach for superior performance and sustainable growth. Moreover, the connection between value innovation and the foundation of valuable, rare, and inimitable resources and capabilities that lead to innovation protection, market dominance, and long-term competitive advantages has not yet been investigated, especially in the Yemeni telecommunications industry.

Furthermore, previous studies on innovation indicate that researchers and telecommunications companies are keenly interested in investigating innovation and its effects on companies' growth and survival [1,2,6,10,19]. The survival and success of telecommunications companies depend greatly on their ability to create new service value that satisfies customers' changing needs [3]. Moreover, the logic of value innovation is seen to represent a new opportunity for telecommunications service providers to break competitiveness, attain sustainable competitive advantages, and assure long-term superior performance [18]. Thus, providing the type of services that are appealing to customers and that can induce post-purchase intentions after using them, leading to increased revenue and effective sustainable development, is a crucial issue for telecommunications service providers [20]. This encourages telecommunications service providers to continuously investigate and improve their service values to meet customers' needs and demands. As telecommunications is essentially a global product, the above observations and the related issues are common in many economies. Telecommunications companies in many countries are now subjected to a global benchmark [19]. For example, being a member of the WTO makes it critically important for Yemen telecommunications companies to be more innovative in order to cope with the resulting market liberalization. Therefore, the survival and success of telecommunications companies, such as those in Yemen, depends greatly on their ability to innovatively identify, develop, protect, and deploy value-creating resources that are likely to be rare, valuable, and imperfectly imitable, thus attaining long-term superior performance and sustainable growth [18,21].

According to Kuo et al. [20], the key to corporate success and competitive advantage is enhancing service quality, perceived value, and customer satisfaction. Only a limited number of research papers have discussed value innovation and its effects on customer satisfaction, customer loyalty, and sustainable growth. Especially in Yemen, no large-scale study has investigated the nature of value innovation in the telecommunications industry. This empirical study aims to provide a perspective, analysis, and strategies for telecommunications service providers in Yemen to promote value innovation successfully. As Yemen recently joined the World Trade Organization (WTO), the telecommunications market of Yemen has been liberalized, which has led to intense competition, particularly between the national and multinational mobile service providers. Therefore, to overcome these existing research gaps, the main aim of this study was to develop and validate a conceptual model concerning value innovation and its impact on a firm's performance and sustainable growth by examining the mediating effect of customer satisfaction and loyalty. The proposed model will examine the relationship between value innovation, customer satisfaction, and customer loyalty and their effect on companies' performance and sustainable growth. Furthermore, this study highlights the connection between value innovation and the foundation of valuable, rare, and inimitable resources and capabilities that lead to innovation protection, market dominance, and a long-term competitive advantage.

Moreover, firms need to be more than creative to survive; they require innovative processes and resource management processes that can achieve enduring superior performance and sustainable growth [3,18,22–24]. It is important to innovatively break through the competition and create one's own market space, yet copying and adapting the innovations of others is common. Thus, the significance of this study is to enhance innovation management through the creation of value innovation that improves customer satisfaction, customer loyalty, and companies' performance and sustainable growth within the context of the RBV of the firm as an underpinning theory. Merging the so-called Blue Ocean principles with the RBV perspective may lead to value innovation, involving the creation of resources with unique, rare and inimitable attributes, which, in consequence, enables companies to achieve sustainable competitive advantages and long-term superior performance. Moreover, value innovation helps in establishing a company's reputation (customer satisfaction) and customer loyalty, which are valuable intangible resources in terms of maintaining a sustainable competitive advantage and superior performance [22,25–27].

This empirical study involves theory development rather than the confirmation of existing theories. We adopted value innovation as an approach towards high customer satisfaction, great customer loyalty, superior performance, and sustainable growth [28]. Furthermore, we investigated the nature of value innovation in the mobile service Industry in the Republic of Yemen. Hence, we formulated the following research questions: First, how does value innovation affect customer satisfaction, customer loyalty, companies' performance, and companies' sustainable growth? Second, how can telecommunications companies break away from the competition, create an uncontested market, and make the competition irrelevant?

The rest of this paper is organized as follows. The theoretical background is presented in Section 2 and the research model and development of the hypotheses are discussed in Section 3. Section 4 discusses the research methodology and data analysis procedures. Then, in Section 4, the study results are presented, and in Section 5, the study findings are discussed. Section 6 discusses the study's theoretical contribution and managerial implications. Finally, in Section 7 we summarize the study's findings and limitations, and provide future recommendations.

## 2. Theoretical Background

### 2.1. Value Innovation

The notion of value innovation aims to provide a leap in value for both customers and firms by enabling business differentiation, reducing competition relevance, and creating uncontested market space [11]. Kim and Mauborgne [12] considered value the key driver of

any innovation success. Without value, innovation appears to be technology-driven, market pioneering, or futuristic, shooting beyond customers' considerations and willingness to pay. From this perspective, Kim and Mauborgne [12] addressed value innovation as the result of a combination of eliminating, reducing, enhancing, and newly creating key elements of products or services.

The term value innovation has been correlated with various perspectives and linked with several analytical tools to assist firms in creating breakthroughs as a strategic move. For instance, Mohanty [29], Mele [30], Mele et al. [31], and Kachouie et al. [32] viewed value innovation from a firm's perspective as involving resource integration and the development of superior competency, whereas Setijono [33] described value innovation with regards to the creation of value for stakeholders. In that study, value innovation was described as having a disruptive-attractive quality, providing a firm with a total solution, extraordinary experiences, and cost reductions through product, service, and delivery platforms [33,34]. Agnihotri [14], Rabino et al. [35], Chang [36], and Kim and Mauborgne [37] considered value innovation a business strategy that should embrace the activities of the entire system of a firm in order to attain a leap in value for customers and a competitive advantage and profitable growth for the firm. In addition, Matthyssens et al. [38], Lindgreen et al. [39], Rønning et al. [40], Faghat et al. [41], Matthyssens [42] examined value innovation from a strategic innovation perspective as the reconceptualization of a business model or industry, the redefinition of a business, or the re-designing of value conceptions or delivery modes, with the ultimate aim of creating new and superior customer value.

Prior studies identified four key drivers of value innovation, namely, culture, processes, people, and resources [43,44]. Mohanty [29] emphasized the achievement of breakpoints in relation to value innovation logic by examining nine elements: "robustness, price, lead time, flexibility, process design, reliability, product design, service empathy, and information system". Simon and Luc [45] examined the impact of systems integration on innovation and customer satisfaction. Moreover, Christa et al. [46] investigated the role of value innovation capabilities in improving company performance in the banking service sector with the influence of the external factors of market orientation and social capital. They concluded that innovation is only partially linked to customer satisfaction, but this result was due to the innovation dimensions (process innovation, organization innovation, and marketing innovation) that they used, which were weakly interlinked to customer satisfaction.

## 2.2. The Resources-Based View (RBV)

The RBV of a firm is a dominant theory in strategic management, explaining the concepts of resources management, competitive advantage, and the superior performance of a firm [26,27,47–49]. The attraction to the RBV comes from the concept of difficult-to-imitate attributes of firms' resources as a source of superior performance and a sustainable competitive advantage [48,50,51]. Barney [21] specified four main attributes required for a firm's resources to gain a sustainable competitive advantage: valuable, imperfectly substitutable, rare, and imperfectly imitable. In this context, prior studies have highlighted intangible resources and capabilities as the key resources for conferring a sustainable competitive advantage that is reflected in superior performance for the firm [18,21,22,25,27,47,52–56]. Moreover, Fahy [54] emphasized the need for intangible resources to drive robust value creation compared to duplicative efforts to gain a sustained competitive advantage. Wang and Lo [26] and Clulow et al. [27] discussed value creation with regards to the customer-focused perspective, explaining the role of a firm's key intangible resources in creating customer value, thereby enhancing customer satisfaction and loyalty and driving customer-related performance. Furthermore, Khan et al. [57] empirically highlighted the significant role of intangible resources and capabilities in enhancing a sustainable competitive advantage and a firm's performance.

### 2.3. Customer Satisfaction

Customer satisfaction has been considered one of the most effective dimensions for superior performance, competitive advantage, and the long-term success of a firm [26,27,58–60]. The RBV literature highlights the notion of superior customer-focused performance, which is achieved through a set of interlinked business processes and the coordination of strategic resources with the goal of satisfying customer needs [26]. Moreover, innovation scholars [61–64] have indicated that innovation is a strong driver of customer satisfaction and firm performance, particularly in service industries. Bellingkrodt and Wallenburg [64] emphasized innovation's key role in increasing the value of delivered services, leading to higher customer satisfaction and loyalty, either by offering new services or enhancing existing services.

Customer satisfaction is a strong predictor for behavioral variables in the telecommunications industry, such as re-purchase intentions, word-of-mouth recommendation, and loyalty [20,65]. Prior studies have also highlighted the importance of customer satisfaction factors for mobile service providers to maintain or improve their market share, customer retention, and profitability [19,20,60,65–69].

### 2.4. Customer Loyalty

Understanding customers and establishing long-term profitable relationships is essential for a firm to enhance sustainability and profitability. RBV and innovation literature emphasize customers' loyalty (brand loyalty) as an intangible valuable resource and a key factor for a firm's sustainable competitive advantage and superior performance [20,26,65,66,68–72]. Diaw and Asare [68] showed a positive relationship between innovation, customer satisfaction, and customer retention in the telecommunications industry. According to Lin and Wang [65], attaining new customers is considerably more expensive than retaining existing customers, which can render many customer relationships unprofitable in the early years. From this perspective, retaining customers through building customer loyalty has become a financial imperative for many companies aiming to win a market share and develop a sustainable competitive advantage [68,69].

### 2.5. Company's Performance

The company's performance refers to the efficiency and effectiveness with which a company can utilize its resources in generating economic outcomes [6]. Asikhia [73] described the measurement of a company's performance based on the company's economic (e.g., profits, sales, return on investment) and strategic (e.g., market share, brand awareness) objectives achieved in the marketplace. Chaudhry et al. [69] emphasized the high importance of innovation capabilities in improving a firm's performance by converting innovative ideas into economic value and profit. Furthermore, Baia et al. [50] highlighted the significance of resources and the rareness of the capabilities of a firm to the creation of a competitive advantage, and thus superior performance. Prior studies [2,3,6,74,75] have confirmed the significant positive effect of innovation on companies' performance, particularly in telecommunications industries.

### 2.6. Sustainable Growth

Corporate sustainability is meant to "meet the needs of a firm's direct and indirect stakeholders, such as shareholders, employees, clients, pressure groups, communities, etc., without compromising its ability to meet the needs of future stakeholders as well" [76]. According to Holliday [77], sustainable growth generates business value for both customers and shareholders as it seeks to make more customers by developing markets that promote and sustain economic prosperity. According to Kim [78], the sustainable growth of a firm is attained by value creation, which depends ultimately upon the evolution of customer needs. In contrast, the power has shifted from companies to customers due to the Internet revolution.

Prior studies have indicated that innovation is a key factor for long-term success and survival [3,6,23,71,74,79]. According to Verma and Singh [71], telecommunications companies must focus on customer satisfaction, which leads to customer loyalty, to achieve sustainable growth.

## 3. Research Model and Development of Hypotheses

Prior scholars have focused on examining value innovation with respect to the innovation process and the value chain [29,41,43,44,80,81]. There is a gap in the literature with regard to empirically linking value innovation with customer satisfaction and loyalty and examining its impact on a firm's superior performance and sustainable growth. In this empirical study, we focus on examining the impact of value innovation on customer satisfaction, customer loyalty, companies' performance, and sustainable growth. Therefore, we investigate the output of value innovation based on three key elements: (1) customer value, (2) shareholder value, and (3) business uniqueness [37,44,82]. According to Kim and Mauborgne [37], "value innovation occurs only when companies align innovation with utility, price, and cost position". To a customer, value innovation is the perceived value in terms of quality, benefit/utility, and price. According to Woodruff [83] "customer value is something customers perceive rather than objectively determined by a seller". It involves a trade-off between the quality, benefits, worth, and utilities that customers receive and the price and sacrifices that they pay or give up. Hence, customer value was examined with regard to four independent dimensions: (1) quality, (2) utility, (3) worth, and (4) price.

The second element examined was shareholder value. According to Wang and Lo [26], shareholders' value is not only measured in financial returns but also in building intangible resources such as a firm's reputation and brand name and increasing investments in R&D, training, and service systems. Furthermore, value innovation emphasizes the cost side of the business model to create a leap in shareholder value in profit [37]. Thus, in this study shareholder value was measured based on four dimensions: (1) cost, (2) profits, (3) performance, and (4) assets/resources. Thirdly, business uniqueness is the premise of the RBV concept of the heterogeneity of resources. According to Tarafdar and Gordon [84], the RBV posits that a firm's value can be determined when unique resources characterize the firm. Barney [21] suggested two relevant features of firm resources necessary to achieve a competitive advantage: (a) these resources must enable the creation of value, and (b) they must resist the duplicative efforts of competitors. According to Barney [21], "firms compete and create value on the basis of resources that are unique, rare, valuable, and not easily imitable or substitutable" [50,84]. Hence, system uniqueness was examined through four independent dimensions: (1) value, (2) rareness, (3) imitability, and (4) sustainability.

Furthermore, previous studies have highlighted the positive effects of RBV and innovation on customer and shareholder value [18,26,27,37,50,85]. They addressed the influence of customer-focused/customer-oriented business principles on customer satisfaction and concluded that both innovation and RBV positively affect customer satisfaction [26,27,58,59,61–64]. Furthermore, previous empirical studies examining telecommunications services have pointed out that perceived value has a strong positive effect on customer satisfaction [20,65,66,86,87].

Ogunnaike et al. [17] examined service innovation in the telecommunications industry and suggested that service innovation significantly affects firms' performance. Furthermore, prior innovation studies have agreed that innovation positively influences companies' performance [2,3,6,23,74,79].

According to Setijono [82], the focus of value innovation goes beyond merely satisfying and delighting existing customers; it is about acquiring new customers in newly created markets. Scholars have also emphasized the necessity of innovation and RBV for firms' survival and sustainable growth [2,6,23,26,27,37,47,48,79,88].

Moreover, prior studies have examined the mobile phone market and found that customer satisfaction has a positive effect on customer loyalty [20,65–68,71]. As this study focuses more on the impact of value innovation and its influence on telecommunications

users, customer satisfaction was viewed as cognitive-based and regarded as an affective state resulting from the cognitive evaluation process [67]. Furthermore, customer satisfaction was treated as a mediator variable and was examined based on two dimensions: (1) needs/demand and (2) expectations. Furthermore, previous researchers examined mobile service industries and suggested customer loyalty to be a key factor for companies' long-term success and viability [20,65,66,68–71]. Chaudhuri and Holbrook [72] suggested two types of loyalty: behavioral (purchase) and attitudinal. According to Lin and Wang [65], "behavioral loyalty consists of repeated purchases of the brand, whereas attitudinal loyalty includes a degree of dispositional commitment in terms of some unique value associated with the brand". As in the work of Lin and Wang [65], in this study we adopted both attitudinal commitment and behavioral re-purchase intention as indicators of loyalty. Hence, customer loyalty was considered as a mediator variable and measured with regards to two dimensions: (1) re-purchase intention/usage continuity, (2) recommendation willingness. Thus, Hypothesis H1 was homogeneously proposed as follows:

**Hypothesis 1 (H1).** *Value innovation has a positive effect on customer satisfaction.*

According to Setijono [82], the focus of value innovation is beyond merely satisfying and delighting existing customers; it is about acquiring new customers in newly created markets. Scholars have emphasized the necessity of innovation and RBV for a firm's survival and sustainable growth [2,6,23,26,27,37,47,48,79,88]. Hence, Hypothesis H2 was proposed as follows:

**Hypothesis 2 (H2).** *Value innovation has a direct positive effect on companies' sustainable growth.*

Furthermore, prior studies have examined the mobile phone market and found that customer satisfaction positively affects customer loyalty [20,65–68,71]. As this study focuses more on the impact of value innovation and its influence on telecommunications users, customer satisfaction was viewed as cognitive-based and regarded as an affective state resulting from the cognitive evaluation process [67]. Furthermore, customer satisfaction was treated as a mediator variable and was examined based on two dimensions: (1) needs/demand and (2) expectations. Thus, Hypothesis H3 was proposed as follows:

**Hypothesis 3 (H3).** *Customer Satisfaction has a positive effect on customer loyalty.*

Furthermore, previous studies have examined mobile service industries and suggested customer loyalty to be a key factor for companies' long-term success and viability [20,65,66,68–71]. Chaudhuri and Holbrook [72] suggested two types of loyalty: behavioral (purchase) and attitudinal. According to Lin and Wang [65], behavioral loyalty consists of repeated purchases of products from a brand. In contrast, attitudinal loyalty includes a degree of dispositional commitment in terms of some unique value associated with the brand. According to the work of Lin and Wang [65], in this study we have adopted both attitudinal commitment and behavioral re-purchase intentions as indicators of loyalty. Hence, customer loyalty was considered as a mediator variable and measured with regard to two dimensions: (1) re-purchase intention/usage continuity and (2) recommendation willingness. As such, Hypothesis H4 and H5 were proposed as follows:

**Hypothesis 4 (H4).** *Customer loyalty has a positive effect on companies' performance.*

**Hypothesis 5 (H5).** *Customer loyalty has a positive effect on companies' sustainable growth.*

Figure 1 depicts the study's conceptual framework. It investigates the effects of value innovation on company performance using customer satisfaction and loyalty as intermediate or extraneous variables. According to the conceptual framework (Figure 1), value innovation was treated as an independent variable and measured using three criteria:

customer value, shareholder value, and business system uniqueness. As shown in the illustration, the impact of value innovation on company performance was measured in two ways: indirectly through increased customer satisfaction and loyalty and directly to achieve long-term sustainable growth. Furthermore, customer satisfaction was measured using the needs and expectations of the customers, whereas customer loyalty was measured using the dimensions of usage continuity and recommendation willingness. Furthermore, we assessed the effects of value innovation on company performance from two perspectives: the short-term response and the long-term response, which is referred to as sustainable growth.

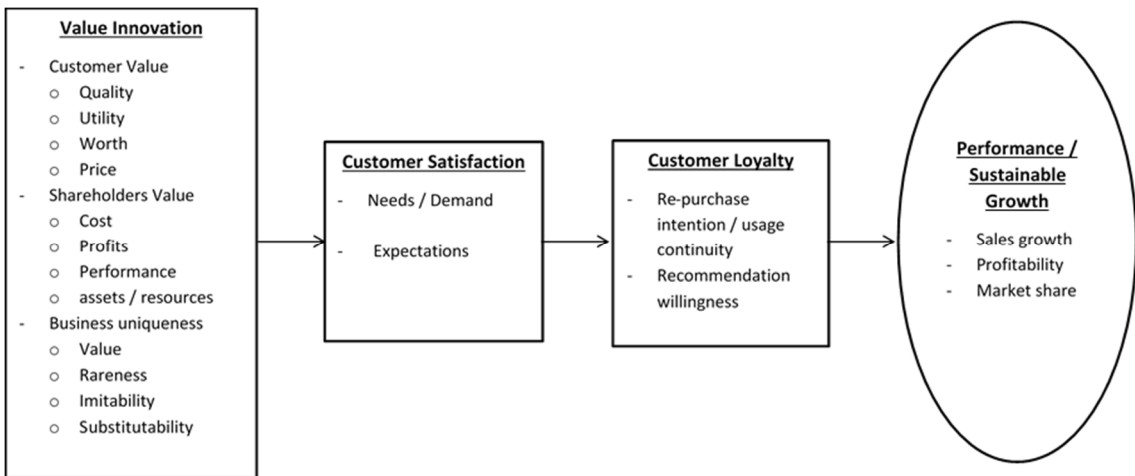

**Figure 1.** Proposed research model.

Ogunnaike et al. [17] examined service innovation in the telecommunications industry and suggested that service innovation significantly affects firms' performance. Furthermore, prior innovation studies have agreed that innovation positively influences companies' performance [2,3,6,23,74,79]. In addition, scholars have emphasized the necessity of innovation and RBV for a firm's survival and sustainable growth [2,6,23,26,27,37,47,48,79,88]. Thus, Hypotheses H6 and H7 were proposed as follows:

**Hypothesis 6 (H6).** *Value innovation has an indirect positive effect on companies' sustainable growth through customer satisfaction and customer loyalty.*

**Hypothesis 7 (H7).** *Value innovation has an indirect positive effect on companies' performance through customer satisfaction and customer loyalty.*

## 4. Methodology

To validate the proposed model and test the hypotheses, we employed a quantitative method to collect data from Yemeni mobile service providers' employees via a paper-based survey. Data were collected sequentially among telecommunications companies and within each telecommunications company department to avoid duplication and missed data. As a result, the convenience non-probability sampling method was adopted to target the respondents, because the likelihood of accessing all employees and having them available at a specific time was limited, especially when using the self-administration collection method [89,90]. The telecommunications industry was chosen for this study due to its highly competitive business environment; the industry's dilemma of market saturation, a high churn rate, and high operational costs; and the critical importance of innovation for sustaining profitable growth; especially for Yemen's telecommunications market, which recently liberalized by joining the World Trade Organization (WTO).

Among 340 distributed questionnaires, 304 were successfully collected, indicating an 89.4% response rate, whereas 36 questionnaires were missed or incomplete. Then,

data screening analysis was conducted to clean and prepare the collected data for further statistical analysis. Therefore, missing values, response patterns, outliers, and the data distribution were examined to ensure the accuracy of data, thus producing a total of 292 questionnaires satisfying the SEM analysis.

To ensure construct validity and reliability, all measurement items were adapted with slight adjustments from various sources among the extant studies. The survey instrument was subjected to content and face validity checks and amendments were made based on the feedback received from the respondents. The items of each factor are presented in Appendix A.

A five-point Likert scale, ranging from 1 (strongly disagree) to 5 (strongly agree), was employed for the items of the factors included in the proposed model. The five-point scale was chosen due to its easy comprehensibility to respondents and the ability to express respondents' views [91], increasing the response rate and response quality and reducing respondents' level of frustration [92]. To increase validity, the instrument was developed based on the research model and by referring to related literature and previous studies of the telecommunications sector in other countries such as China, Korea, Nigeria, Somalia, and the Netherlands [3,6,23,74,79]. A pre-test was conducted by two professors of strategic management and three experts in the telecommunications industry to ensure the validity of the questionnaire context. The pre-test was designed to validate the questionnaire items in terms of face validity, with a focus on the survey's theoretical and practical design. The experts also looked into the terms' suitability for measuring the specific constructs. Moreover, the questionnaire instrument was further validated using a pilot test of 34 responding employees from different management levels and departments in the Yemen Mobile company. The pilot study confirmed the scale reliability of the items by obtaining Cronbach's alpha (*a*) values ranging between 0.819 and 0.953.

The profiles of the respondents revealed that 206 (70.5%) were male and 86 were female (29.5%). The age-range of the majority of mobile telecommunications employees was between 26 and 45 years old (78.1%), followed by the age range between 18 and 25 years old (12.3%), above 46 years old (9.2%), and below 18 years old (0.3%). Aside from that, the vast majority of respondents (72.9%) had bachelor's degrees, whereas 21.5% had higher educational qualifications such as master's and doctoral degrees, and 5.4% had either a diploma degree (4.1) or a high school education and below (1.3%). When the survey was distributed to employees of Yemen's four mobile service providers, 28.4% were from Yemen Mobile Company, 33.2% from MTN Company, 25.3% from Sabafon Company, and 13% from Y-Telecom Company. Similarly, 28.4% of respondents were MTN, Sabafon, or Y-Telecom subscribers, whereas 33.2%, 25.3%, and 13% were MTN, Sabafon, or Y-Telecom subscribers, respectively. Similarly to [93], the nonresponse bias was analyzed by examining the differences between all key variables used in the research model for the respondents in the early and late waves of data collection using the independent *t*-test. No significant differences were detected between the early and late respondent groups, suggesting that there was not a significant response bias.

The survey data collected for this study were analyzed using the Statistical Package for the Social Sciences (SPSS) software and partial least squares structural equation modeling (PLS-SEM). SPSS version 25 was used for screening and assessing the collected data in terms of missing values, outliers, and normality issues. PLS-SEM was employed and SmartPLS software was used to analyze the data by assessing the measurement model and the structural model. PLS-SEM was chosen due to its ability and efficiency in developing complex path models with direct and indirect effects compared to the multiple regression and linear regression approaches [94].

## 5. Results

### 5.1. Common Method Bias

In the present study, we adopted a full collinearity assessment approach to examine the common method bias in PLS-SEM [95]. The full collinearity test is assessed with variance

inflation factors (VIF) for all latent variables in a model, with a VIF value greater than 3.3 indicating pathological collinearity and thus the existence of a possible standard method bias [95]. In the proposed model, the VIF values of all latent variables were less than 3.3, indicating the absence of common method bias contamination affecting the model. Thus, there were no common method bias issues in the measurement model.

### 5.2. Measurement Model Assessment

The measurement model evaluation was conducted to determine the reliability and validity of the construct based on the theoretical context. Table 1 shows the Cronbach's alpha and composite reliability values of all constructs, ranging from 0.850 to 0.954 and 0.909 and 0.959, respectively, above the recommended cut-off values of 0.70 [96]. Therefore, the findings indicate that all constructs demonstrated the strong scale reliability of the measurement model. Furthermore, the results showed high individual-indicator reliability, with all items having factor loading values greater than 0.708 [96]. Similar threshold values were adopted in previous studies (e.g., [15,16,97–103]). Therefore, the convergent validity of the model was confirmed.

**Table 1.** Factor loadings, Cronbach's alpha, CR, and AVE.

| Construct/Item | Factor Loadings | Cronbach's Alpha | CR | AVE |
|---|---|---|---|---|
| Value Innovation (VI) * | | 0.954 | 0.959 | 0.537 |
| Customer Value (CV) | | 0.871 | 0.907 | 0.662 |
| CV1 | 0.837 | | | |
| CV2 | 0.860 | | | |
| CV3 | 0.837 | | | |
| CV4 | 0.800 | | | |
| CV5 | 0.729 | | | |
| Shareholder Value (SV) | | 0.874 | 0.909 | 0.666 |
| SV1 | 0.792 | | | |
| SV2 | 0.877 | | | |
| SV3 | 0.831 | | | |
| SV4 | 0.761 | | | |
| SV5 | 0.816 | | | |
| Business Uniqueness to Customer (BUC) | | 0.870 | 0.906 | 0.658 |
| BU1 | 0.810 | | | |
| BU2 | 0.830 | | | |
| BU3 | 0.795 | | | |
| BU4 | 0.803 | | | |
| BU5 | 0.818 | | | |
| BU6 | 0.823 | | | |
| BU7 | 0.836 | | | |
| BU8 | 0.836 | | | |
| BU9 | 0.836 | | | |
| BU10 | 0.851 | | | |
| Customer Satisfaction (CS) | | 0.850 | 0.930 | 0.869 |
| CS1 | 0.939 | | | |
| CS2 | 0.925 | | | |
| Customer Loyalty (CL) | | 0.894 | 0.950 | 0.904 |
| CL1 | 0.952 | | | |
| CL2 | 0.949 | | | |
| Company's Performance (CP) | | 0.901 | 0.938 | 0.835 |
| CP1 | 0.913 | | | |
| CP2 | 0.925 | | | |
| CP3 | 0.903 | | | |
| Sustainable Growth (SG) | | 0.856 | 0.913 | 0.777 |
| SG1 | 0.897 | | | |
| SG2 | 0.895 | | | |
| SG3 | 0.852 | | | |

* Second-order constructs.

Secondly, the discriminant validity was further evaluated by employing the heterotrait–monotrait ratio (HTMT) to explain the extent to which any two constructs were truly correlated [96]. As shown in Table 2, the HTMT assessment resulted in correlation values below the cutoff (0.90) [104,105], indicating that each construct was more closely related to its indicators, thereby satisfying the discriminant validity criterion.

**Table 2.** Assessment of discriminant validity (HTMT).

| | Company's Performance | Customer Loyalty | Customer Satisfaction | Sustainable Growth | Value Innovation |
|---|---|---|---|---|---|
| Company's Performance | | | | | |
| Customer Loyalty | 0.734 | | | | |
| Customer Satisfaction | 0.658 | 0.888 | | | |
| Sustainable Growth | 0.766 | 0.882 | 0.871 | | |
| Value Innovation | 0.682 | 0.885 | 0.862 | 0.787 | |

### 5.3. Structure Model Assessment

To assess the structural model, path coefficients and *t*-values were used to examine the strength and significance of the relationships between variables in the proposed structural model. The obtained results (Table 3) show that the path coefficients were within the standardized values of 0.247 and 0.779. The *t*-values ranged between 4.496 and 29.283, indicating that the model path coefficients were strongly significant [96]. Moreover, the findings imply a strong positive relationship between independent and dependent variables for all hypotheses, except H2, as they had a positive path coefficient value greater than 0.5. Furthermore, the obtained *p*-values were less than 0.001, confirming the statistical significance of the structural model at the 1% level [96].

**Table 3.** Results of the structural model.

| Hyp. | Path | β | Standard Deviation | *t*-Values | *p* Values | Decision |
|---|---|---|---|---|---|---|
| H1 | Value Innovation → Customer Satisfaction | 0.779 | 0.027 | 29.283 | 0.000 | Supported ** |
| H2 | Value Innovation → Sustainable Growth | 0.247 | 0.055 | 4.496 | 0.000 | Supported ** |
| H3 | Customer Satisfaction → Customer Loyalty | 0.776 | 0.029 | 26.742 | 0.000 | Supported ** |
| H4 | Customer Loyalty → Company's Performance | 0.659 | 0.040 | 16.592 | 0.000 | Supported ** |
| H5 | Customer Loyalty → Sustainable Growth | 0.570 | 0.059 | 9.673 | 0.000 | Supported ** |

Significant at ** $p \leq 0.001$.

As shown in Table 3, the results showed that all direct hypothesized relationships were strongly significant. For instance, the results implied a strong interactive relationship between value innovation and customer satisfaction (H1: β = 0.779, *t* = 29.283, *p* < 0.001), and a positive direct effect of value innovation on companies' sustainable growth (H2: β = 0.247, *t* = 4.496, *p* < 0.001). The results confirmed the strong positive relationship between customer satisfaction and customer loyalty (H3: β = 0.776, *t* = 26.742, *p* < 0.001), with customer satisfaction tending to be a strong predictor of customer loyalty as an increase in customer satisfaction by 1 standard deviation led to a positive increase in customer loyalty by 0.776 units. Furthermore, our findings revealed the positive impact of customer loyalty on companies' performance (H4: β = 0.659, *t* = 16.592, *p* < 0.001) and sustainable growth (H5: β = 0.570, *t* = 9.673, *p* < 0.001), implying that customer loyalty tended to be a strong indicator of companies' performance and sustainable growth.

### 5.4. Assessment of Mediation

To evaluate the mediation effect in the proposed model, the bootstrap analysis method of Preacher and Hayes [106] was employed as a more recent and statistically powerful approach [96]. As shown in Table 4, the mediation test results demonstrated that hypotheses H6 and H7 were supported, in which both indirect effects were significant (H6: β = 0.398, *t* = 9.959, *p* < 0.001 and H7: β = 0.345, *t* = 8.835, *p* < 0.001) and neither of the 95% bootstrapped confidence intervals included zero. Therefore, the findings enabled us to conclude that there was a mediating effect of customer satisfaction and customer loyalty on both companies' performance and sustainable growth.

**Table 4.** Assessment of mediation effect.

| Hyp. | Relationship | Path a | Path b | Path c | Indirect Effect | Std. Dev. | *t*-Values | *p*-Values | 95% LL | 95% UL | Decision |
|------|-------------|--------|--------|--------|-----------------|-----------|-----------|-----------|--------|--------|----------|
| H6 | VI → CS → CL → CP | 0.779 | 0.776 | 0.659 | 0.398 | 0.040 | 9.959 | 0.000 | 0.320 | 0.477 | YES |
| H7 | VI → CS → CL → SG | 0.779 | 0.776 | 0.570 | 0.345 | 0.039 | 8.835 | 0.000 | 0.268 | 0.421 | YES |

## 6. Discussions and Study Contribution

The research model of value innovation adopted in this study showed that value innovation can be an efficient approach by which a firm can obtain superior performance and sustainable growth through providing dynamic business differentiation and superior customer value, which can build up customer satisfaction (reputation) and customer loyalty (brand loyalty). The research model was empirically validated within the telecommunications industry and demonstrated the significant positive effect of value innovation on companies' performance and sustainable growth through enhancing customer satisfaction and loyalty as a mediating factor. Furthermore, the hypothesis testing confirmed the direct significant positive influence of value innovation on companies' sustainable growth. Moreover, for the research model of value innovation, we adopted the RBV with regard to the firm. Thus, the significance of the value innovation concept in this study went beyond providing a quantum leap in value but also played a major role in creating intangible resources, such as knowledge, skills, experience, customer satisfaction, and loyalty. The study highlighted the efficient and effective impact of those intangible resources in creating valuable, rare, and difficult-to-imitate resources and competencies in achieving business uniqueness and thus a sustainable competitive advantage and growth. In this context, the dynamic process of value innovation leads to the creation of more intangible valuable resources, superior performance, a sustainable competitive advantage, and long-term success.

These research results are consistent with the findings of previous studies. The majority of peer-reviewed articles have viewed value innovation as an effective strategy for achieving superior performance, a competitive advantage, and sustainable growth, and this has a positive impact on firm performance, profitability, and growth [107–113]. Furthermore, consistently with earlier studies which validated the positive effect of innovation on a firm's long-term success and sustainable growth [3,6,23,26,27,37,47,48,74,79,88,108,113–117], the results of this study enabled us to conclude that there is a direct and indirect effect of value innovation on sustainable growth. Overall, the findings of this empirical study, in line with the major trends of the literature on value innovation, demonstrated that the value innovation approach could result in superior performance, a competitive advantage, or sustainable growth.

### 6.1. Theoretical Contribution

From the theoretical perspective, this empirical study offers several significant contributions. Firstly, it extends the literature on strategic management by linking the innovation management literature and RBV and marketing-based research. In particular, this study contributes significantly to bridging the gap between value innovation, customer satisfaction, and customer loyalty and their effect on companies' performance and sustainable growth. In other words, in this empirical study we developed a theoretical research model that seeks to drive value innovation, improve customer satisfaction, and increase customer loyalty as a strategy for obtaining superior performance and long-term growth, as well as to connect value innovation to the foundation of valuable, rare, and imitable resources and capabilities, which leads to innovation protection, market dominance, and long-term competitive advantages.

The findings of this paper prove the significant positive relationship between value innovation and customer satisfaction, customer loyalty, company performance, and sustainable growth. Therefore, this study contributed to the value innovation literature by

validating the capability of value innovation to be an approach for long-term superior performance and sustainable growth

Furthermore, the findings of this paper contribute to the continued improvement of the robust logic of value innovation by boosting value innovation protection through the creation of valuable intangible resources such as the firm's reputation (customer satisfaction) and brand loyalty (customer loyalty). As a result, the study provides an excellent perspective on mitigating the imperfection of value innovation imitation by combining emerging "Blue Ocean" principles with the RBV perspective and marketing concepts, resulting in the creation of difficult-to-imitate value innovation that can achieve long-term superior performance and sustainable competitive advantages.

The study represents the first attempt to empirically study value innovation in the Yemeni telecommunications sector, which could motivate further researchers to investigate other business factors and industries.

*6.2. Managerial Implications*

The core of this study has practical significance as it was designed to mitigate industrial dilemmas through theoretical means. This study contributed to practice by identifying the effectiveness of value innovation in overcoming the telecommunications industry's dilemma of intense competition and subscriber growth saturation by enhancing value innovation, which is defined, developed, protected, and deployed with rare, valuable, and difficult-to-imitate based resources. The study provides companies with a broader perspective to attain superior performance, competitive advantages, and sustainable growth.

Moreover, this study contributes to practice by encouraging companies to avoid aggressive competition by focusing on breaking through the competition, creating a leap in value, making the competition irrelevant, and creating new market spaces. In that context, the findings of this study highlight the significance of value innovation in creating a competitive advantage, enhancing quality and productivity, promoting customer satisfaction and loyalty, and hence boosting profitability.

Furthermore, the developed research model provides a better perspective for firms to protect their value innovations and create sustainable competitive advantages through creating intangible resources and capabilities. As a result, we strongly advise telecommunications companies to shift their business strategies from tangible physical asset-focused to intangible asset-focused innovation through value breakthroughs in order to achieve a quantum leap in value, improve customer value, enhance customer satisfaction, and promote customer loyalty, which will ultimately result in superior performance, a competitive advantage, and sustainability.

Moreover, this paper will enrich firms' strategic thinking with the logic of value innovation, with a focus on intangible resources, such as knowledge, skills, the firm's reputation (customer satisfaction), and brand loyalty (customer loyalty) in order to achieve a sustainable competitive advantage and long-term success.

Overall, this empirical study significantly contributes to practice by providing a framework for decision-makers to articulate strategic thinking regarding future market directions toward sustainable success and growth and to make strategic decisions.

## 7. Conclusions, Limitations, and Future Work

Innovation is viewed as critical to a company's survival and growth, which may eventually lead to prosperity and wealth. However, as the market becomes more mature and global, innovation per se becomes more sophisticated, forcing telecommunications companies to shift their innovation strategies in order to improve their competitive position and adapt to the market's rapid dynamic changes. In this regard, based on the perspective of the Blue Ocean strategy, the RBV of the firm, and related marketing theories, in this study we developed a theoretical model to correlate and explain the relationship between value innovation and customer satisfaction, customer loyalty, company performance, and sustainable growth. A questionnaire survey and SEM statistical analysis were used in

conjunction with SmartPLS and SPSS 23 software to examine the proposed research model and test the interrelationships of the hypotheses. As a result, the data analysis included 292 employees of Yemeni mobile telecommunications service providers. Furthermore, the results demonstrated that all indicators of the measurement model and the structural model had significant reliability and validity. The findings indicated that the structural model had a high statistical significance (*p*-value 0.001) and positive relationships between independent and dependent variables for all hypotheses. The study's findings confirmed the existence of a mediation effect of customer satisfaction and loyalty on both companies' performance and long-term growth. Furthermore, this study demonstrated the importance of value innovation in creating intangible resources such as knowledge, skills, experience, customer satisfaction, and loyalty, all of which are critical in achieving a long-term competitive advantage. Finally, the logic of value innovation can be an efficient approach for firms aiming to achieve superior performance, long-term success, and a sustainable competitive advantage.

## 7.1. Limitations

This study, like any other, has limitations. The study was solely concerned with studying innovation management and its contributions to the communications industry; as a result, less attention was paid to other related or innovative industries. Electronics, information technology, and communications industries, for example, are associated with innovation and market development. Although the proposed model is applicable to studying the effects of value innovation in general, the data collection process was limited to mobile telecommunications service providers. As a result, there was a scarcity of providers of the public switched telephone network (PSTN), the internet, and MVAS services. Furthermore, third-party companies such as vendors, suppliers, and contractors were not included in the data collection process. Another limitation discovered while conducting this thesis was the scarcity of secondary data on Yemen's telecommunications industry, specifically innovation management practices. Telecommunications companies in Yemen have strict security and confidentiality data policies, making obtaining reports and documents on innovation practices difficult. In addition to these, one other limitation regarding the model validation was based on the application of an instrument based on perception. Therefore, future researchers are encouraged to measure the proposed model using data based on specific indicators such as the drop-out rate (the number of customers who stopped using the service after particular periods, before or after the expiration of the contract, divided by the total number of customers), the retention index (the number of customers who renewed the contract, divided by the total number of customers), etc.

## 7.2. Future Recommendations

As the telecommunications business environment rapidly moves toward globalization and trade wars, not only between companies but also between economically rich and powerful countries, the need for a dynamic value innovation process that creates business differentiation, low costs, and long-term competitive advantages has become critical for firm success and survival. The majority of existing research has concentrated on the factors associated with the adoption and implementation of value innovation. In contrast, less emphasis has been placed on the long-term impact of value innovation on company performance and long-term growth. As a result, further research into the long-term effects of value innovation is highly recommended. Further research is also recommended into other factors that may affect value innovation in customer-focused prospects, such as customer trust, culture, and environment.

Furthermore, researchers are encouraged to thoroughly investigate the phenomenon of the digital transformation of the communications market, which has the potential to create new market spaces while eliminating others, to analyze new business opportunities, risks, and challenges. Furthermore, scholars are encouraged to explore value innovation in various industries. Future researchers, for example, are advised to test the research

model proposed in this thesis with different industries and analysis methods to generate interesting results.

**Author Contributions:** Formal analysis, M.A.H. and M.A.A.-S.; Supervision, D.N.I. and M.R.D.; Writing—original draft, M.A.H. and M.A.A.-S.; Writing—review & editing, M.A.H., M.A.A.-S. and N.A.I.; funding acquisition, A.A.A., G.A. and S.K.T. All authors have read and agreed to the published version of the manuscript.

**Funding:** This research was funded by Centre of excellence from iRMC of Universiti Tenaga Nasional with grant number J510050002-IC-6 BOLDREFRESH2025-CENTRE OF EXCELLENCE.

**Institutional Review Board Statement:** Not applicable.

**Informed Consent Statement:** Not applicable.

**Acknowledgments:** The authors would like to acknowledge the support for this publication provided through J510050002-IC-6 BOLDREFRESH2025-CENTRE OF EXCELLENCE from the iRMC of Universiti Tenaga Nasional.

**Conflicts of Interest:** The authors declare no conflict of interest.

## Appendix A. Constructs, Items, and Their Sources

**Table A1.** Constructs, Items, and Their Sources.

| Code | Question | Source |
|---|---|---|
| | Customer Value (CV) | |
| CV1 | The telecom services that I am using are reliable | |
| CV2 | The telecom service that I am using is useful and fulfils my needs | [66,118] |
| CV3 | My telecom service provider is sincere and patient in solving my problems | |
| | Shareholder Value (SV) | |
| SV1 | My company offers/maintains a telecom service that has an efficient cost structure. | |
| SV2 | My company maintains and improves a profitable telecom service. | |
| SV3 | My company maintains and improves telecom service that shows interest to customers. | [26,37,44] |
| SV4 | My company focuses on telecom service which is associated with the unique assets (skills, resources, and capabilities) of the company. | |
| SV5 | My company focuses on telecom service which improves shareholder value. | |
| | Business Uniqueness to Customer (BUC) | |
| BU1 | My telecom service offers more value and capabilities than its competitors. | |
| BU2 | My telecom service's rare and unique resources and capabilities are more attractive than those of its competitors. | |
| BU3 | My telecom service's inimitable and difficult-to-copy resources and capabilities are more preferred than those of its competitors. | [21,27] |
| BU4 | My telecom service's sustainable resources and capabilities are more preferred than those of its competitors. | |
| BU5 | My telecom service's unique attributes are more attractive than those of its competitors. | |
| BU6 | My company attempts to offer/maintain a telecom service that has valuable resources and capabilities. | |
| BU7 | My company attempts to offer/maintain a telecom service that has rare and unique resources and capabilities. | [21,27] |
| BU8 | My company attempts to offer/maintain a telecom service that has inimitable and difficult-to-copy resources and capabilities. | |

**Table A1.** *Cont.*

| Code | Question | Source |
|------|----------|--------|
| BU9 | My company attempts to offer/maintain a telecom service that has sustainable resources and capabilities. | |
| BU10 | My company attempts to offer/maintain a telecom service that has unique attributes. | |
| | Customer Satisfaction | |
| CS1 | The telecom service that I am using satisfies my needs. | [26,65] |
| CS2 | The telecom service that I am using meets my expectations. | |
| | Customer Loyalty | |
| CL1 | I am willing to re-purchase or continue using the telecom service that I am using. | [65,119] |
| CL2 | I am willing to recommend telecom services that I am using to other people. | |
| | Company Performance | |
| CP1 | My company's sales growth has increased in the last two years. | |
| CP2 | My company's profits have increased in the last two years. | [6,17] |
| CP3 | My company's market share has increased in the last two years. | |
| | Sustainable Growth | |
| SG1 | My company's sales growth is growing and expected to grow. | |
| SG2 | My company's profit is growing and expected to grow. | [6,74] |
| SG3 | My company's market share is growing and expected to grow. | |

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
