# Peer review of "The Effect of Value Innovation in the Superior Performance and Sustainable Growth of Telecommunications Sector: Mediation Effect of Customer Satisfaction and Loyalty"

_sustainability, doi:10.3390/su14106342_

Round 1

Reviewer 1 Report

I read the article with great interest. Indeed, value innovation is a relatively new topic, not fully researched and still controversial, similar to previous research on intellectual capital, especially the relational capital that is part of it. 
I rate the article highly due to its good, supported by a thorough review of scientific literature, grounding in literature. Moreover, the research concept itself and the way of conducting the proof I also rate highly. A good choice of methods, adequate to the research questions posed, is more than half of success in scientific work. 
The list of cited literature sources is impressive. I notice self-citations, but they are correct and do not exceed acceptable limits.  I wish Authors the best of luck and the implementation of the research planned for the future. 

Author Response

 Dear Reviewer 1

Thank you for giving us the opportunity to submit a revised draft of the manuscript " The Effect of Value Innovation in the Superior Performance and Sustainable Growth of Telecommunications Sector: Mediation Effect of Customer Satisfaction and Loyalty " for publication in the Sustainability. We appreciate the time and effort you dedicated to providing feedback on our manuscript and are grateful for the insightful comments and valuable improvements to our paper. We have highlighted the changes within the manuscript. Here is a point-by-point response to the reviewers' comments and concerns.

Thank you, and we highly appreciate your kind consideration and cooperation in approving and listing our manuscript for publication shortly. Please see the attachment for details of the revised manuscript.

Comments and Suggestions for Authors

I read the article with great interest. Indeed, value innovation is a relatively new topic, not fully researched and still controversial, similar to previous research on intellectual capital, especially the relational capital that is part of it. I rate the article highly due to its good, supported by a thorough review of scientific literature, grounding in literature. Moreover, the research concept itself and the way of conducting the proof I also rate highly. A good choice of methods, adequate to the research questions posed, is more than half of success in scientific work. The list of cited literature sources is impressive. I notice self-citations, but they are correct and do not exceed acceptable limits.  I wish Authors the best of luck and the implementation of the research planned for the future.

Response

Thank you for the positive feedback.

Reviewer 2 Report

Ad. 1

As the title suggests, the problem explored in this paper is the effect of value innovation on superior performance and sustainable growth. A comprehensive review of similar research is presented in the literature review. Different types of values ​​and different types of their innovations are mentioned. Although not specifically mentioned, innovation varies from evolutionary to radical revolutionary.

It is not clear what value innovations were specifically the subject of this research. The context of the research is not clear. Innovation can vary from evolutionary to radical revolutionary. The authors should determine what kind of innovations they take into account in the research. Is this research after a particular innovation cycle (eg. innovation of the end-user delivery system, or innovation of the value of the CRM program used by companies internally), so that it is clear to all respondents what value innovation is, or did the questions relate in general to any value innovation?

If the latter is the case, I fear the hypotheses are trivial because the obvious claim is that without innovation (and even value innovation) business models and companies are not sustainable, and that innovation (generally) allows survival and growth, although it does not guarantee it. This is almost axiomatic in economic theories and management.

Therefore, it is necessary for the authors to state exactly what innovations were the subject of research, and whether all respondents had the same experience and understanding of innovation.

Ad. 2

The instrument, ie the questionnaire and the constructs should be described because the entire research is based on it. It is necessary to describe the methodology of instrument design and validation.

Ad. 3

H1 (284 to 287) is obviously wrong, that is, there was an error.

Ad. 4

As stated in the paper (311 to 315), "researcher employed a quantitative method to collect data from Yemeni mobile service providers' employees via a paper-based survey. Data was collected sequentially among telecommunications companies and within each telecommunications company department to avoid duplication and missed collection. " From the description of the methodology, it is clear that the instrument is based on the perception of the respondents, and not on the measurement of some process variables or financial indicators. This approach has a sense in some indicators, such as respondents’ frustration level (333). However, research into the impact of value innovation on some of the exact indicators listed in the hypotheses (288 to 295) would be better based on insight into actual data than on respondents ’ perceptions. This applies, for example, to H2, the impact of Value innovation on companies ’sustainable growth (288). Company growth data is more obvious and accurate in financial statements than in respondents ’ perception. The same is true with the H3 which deals with customer loyalty. Customer loyalty is exactly measured as retention rate, drop-out rate, etc. It is exactly the same with H4 companies’ performance, which is actually a complex indicator, composed of several elementary indicators in the field of finance, sales, human resources, etc..

Therefore, the best validation of constructs and instruments as used in this research is in relation to objective reality, ie data that can be accurately measured. Such validation also provides an answer as to whether the respondents’ perception of a phenomenon is consistent with objective reality, which is not always the case for a variety of reasons. The authors need to supplement the research in this regard.

Author Response

Dear Reviewer 2

Thank you for giving us the opportunity to submit a revised draft of the manuscript " The Effect of Value Innovation in the Superior Performance and Sustainable Growth of Telecommunications Sector: Mediation Effect of Customer Satisfaction and Loyalty " for publication in the Sustainability. We appreciate the time and effort you dedicated to providing feedback on our manuscript and are grateful for the insightful comments and valuable improvements to our paper. We have highlighted the changes within the manuscript. Here is a point-by-point response to the reviewers' comments and concerns.

Thank you, and we highly appreciate your kind consideration and cooperation in approving and listing our manuscript for publication shortly. Please see the attachment for details of the revised manuscript.

Comments and Suggestions for Authors

Ad. 1

As the title suggests, the problem explored in this paper is the effect of value innovation on superior performance and sustainable growth. A comprehensive review of similar research is presented in the literature review. Different types of values ​​and different types of their innovations are mentioned. Although not specifically mentioned, innovation varies from evolutionary to radical revolutionary. It is not clear what value innovations were specifically the subject of this research. The context of the research is not clear. Innovation can vary from evolutionary to radical revolutionary. The authors should determine what kind of innovations they take into account in the research. Is this research after a particular innovation cycle (eg. innovation of the end-user delivery system, or innovation of the value of the CRM program used by companies internally), so that it is clear to all respondents what value innovation is, or did the questions relate in general to any value innovation? If the latter is the case, I fear the hypotheses are trivial because the obvious claim is that without innovation (and even value innovation) business models and companies are not sustainable and that innovation (generally) allows survival and growth, although it does not guarantee it. This is almost axiomatic in economic theories and management. Therefore, it is necessary for the authors to state exactly what innovations were the subject of research and whether all respondents had the same experience and understanding of innovation.

Response:

Thank you for the valuable comments. We agree with your comments, and we have improved the introduction section by adding the exact definition of value innovation and its type in the revised version of this manuscript. Please refer to the introduction section [lines 63 – 70].

Ad. 2

The instrument, i.e., the questionnaire and the constructs should be described because the entire research is based on it. It is necessary to describe the methodology of instrument design and validation.

Response:

Thank you for highlighting this point. We agree with the reviewer concerning this comment. We have added the adapted constructs, items, and sources along with the instrument design and validation; please refer to section 4 [lines 341 – 344] and Appendix 1 in the revised version of the manuscript.

Ad. 3

H1 (284 to 287) is obviously wrong, that is, there was an error.

Response:

Thank you for highlighting this; the typo mistakes have been corrected in the revised version of the manuscript.

Ad. 4

As stated in the paper (311 to 315), "researcher employed a quantitative method to collect data from Yemeni mobile service providers' employees via a paper-based survey. Data was collected sequentially among telecommunications companies and within each telecommunications company department to avoid duplication and missed collection. " From the description of the methodology, it is clear that the instrument is based on the perception of the respondents, and not on the measurement of some process variables or financial indicators. This approach has a sense in some indicators, such as respondents' frustration level (333). However, research into the impact of value innovation on some of the exact indicators listed in the hypotheses (288 to 295) would be better based on insight into actual data than on respondents' perceptions. This applies, for example, to H2, the impact of Value innovation on companies' sustainable growth (288). Company growth data is more obvious and accurate in financial statements than in respondents' perceptions. The same is true with the H3 which deals with customer loyalty. Customer loyalty is exactly measured as retention rate, drop-out rate, etc. It is exactly the same with H4 companies' performance, which is actually a complex indicator, composed of several elementary indicators in the field of finance, sales, human resources, etc. Therefore, the best validation of constructs and instruments as used in this research is in relation to objective reality, ie data that can be accurately measured. Such validation also provides an answer as to whether the respondents' perception of a phenomenon is consistent with objective reality, which is not always the case for a variety of reasons. The authors need to supplement the research in this regard.

Response:

We agree with the reviewer's comment that the instrument is based on the respondents' perceptions. As we mentioned in the revised version of the manuscript, all measurement items were adapted with slight adjustments from well-known sources of the extant studies in the innovation value. In addition, the proposed model was verified based on the real responses of companies' employees and managers who are more familiar with their companies. Therefore, the measurement items have been added to the revised version of the manuscript.

Reviewer 3 Report

  1. Without a strong engagement in recent studies, the study seems generic and cursory, the arguments seem patched and do not extend much beyond existing literature, and the paper does not seem to provide the type of deep, novel theoretical contributions we typically seek at novel research. Prior literature reviewed in your study looks outdated. You need to strengthen the section with literature/references based on the articles published recently, to convince that your paper examines prior work and has some new/novel paradigm.
  2. There must be some reasons or explanations for why testing this model is important in Yemen. The linkage between the arguments is either missed or weak However, why do you have to test the impact of value innovation on firm performance and long-term growth by examining the mediation effect 24 of customer satisfaction and loyalty.? What kind of Companies are you talking about? Is it global Companies and what is the context? There is no explanation about the industry of in the introduction. I think this should be revised.
  3. The introduction contains much general information frequently in more than one section and there is no preamble to the research and its importance what do you want to do.
    A further effort is needed in the introductory section, for instance, there is a need to give a general overview of your study knowing fully well that your audience is expanded; also the latter part of the introduction should have the statement of the problem, research question, and objective, in order.
    - The research question is not clear. Make strong discussions about why the research is important, what motivations are required to complete the research and what gaps you want to search?
    -I could not understand what are the gaps or contributions of this research. In gap and contribution part makes the case more strongly by depending on the recent literature review about the topic and what is the needed gaps. What was the key contribution(s) of this study that would not have been known from existing research? The arguments should go beyond "few studies investigated these effects, and therefore we wanted to investigate it". For example, "Only a limited number of research papers” " no large-scale study investigates the nature of value innovation in the telecommunications industry.". “and long-term competitive advantage is not investigated yet”. Given These arguments are not sufficient. Even without the evidence from this research, the relationships among study constructs were straightforward and intuitive. The authors should make salient and strengthen this study's unique contributions in the introduction and discussion sections.
  4. Literature Review-This is bulky with some irrelevant information; it should be summarized under two to three pages. A well-summarized review of related literature would open the glaring gap and give relevance to the intending study. Make the case more strongly in the literature review as to why you're doing this research. It could be much stronger as you can point to truthfulness as an important construct. The paper could use that. You need to strengthen the section with literature/references based on the articles published recently, to convince that your paper
  5. you need to enlarge the literature background and references. Some authors are cited many times in the document and support several affirmations and hypotheses with the same references all over again. Please update references with the latest developments in the sector and include more varied sources of knowledge. I am including some references that might help, this does not mean you necessary include them all in the manuscript:
    1. Ibrahim, B. 2021. “Social Media Marketing Activities and Brand Loyalty: A Meta-Analysis Examination.” Journal of Promotion Management. 1–31. doi:10.1080/10496491.2021.1955080
    2. Ibrahim, B., & Aljarah, A. (2021). The era of Instagram expansion: matching social media marketing activities and brand loyalty through customer relationship quality. Journal of Marketing Communications, 1-25.
    3. Ting, H., K. Lim Tan, L. Xin Jean, and C. Jun Hwa. 2020. “What Determines Customers’ Loyalty Towards Telecommunication Service Mediating Roles of Satisfaction and Trust Responsible Tourism View Project Marketing View Project.” Article in International Journal of Services Economics and Management. doi:10.1504/IJSEM.2020.111179.Jacoby, J., and R. W. Chestnut. 1978. Brand Loyalty : Measurement and Management. New York, NY: Wiley.
    4. Chaudhuri, A., and M. Holbrook. 2001. “The Chain of Effects from Brand Trust and Brand Affect to Brand Performance: The Role of Brand Loyalty.” Journal of Marketing 65 (2): 81–93. doi:10.1509/ jmkg.65.2.81.18255
  6. The theoretical framework also needs strengthening. the innovation of the paper is not enough. For example, what did we learn about Value innovation and The Resources-Based View (RBV)? It is unclear how the theoretical framework really influenced this study. it isn’t used to rationalize any Hs until H7.
  7. Each hypothesis from H1 to H7 should discuss separately with strong arguments
  8. The screening procedure is explained somewhat convolutely. Can the authors walk the

readers through the procedure? This is relevant not only to understand how the study ended up with a certain sample size but also to appreciate the selection that the sample went through. From this section, it's also unclear whether all respondents are reflecting upon the same brand pages, different brand pages, etc.

  1. Findings: The authors should combine the findings and arguments from previous literature appropriately.  The objective is to develop a clear and convincing text, in which the hypotheses they formulate are the logical conclusions of the storyline of the paper. I cannot capture any further key information regarding variables and measures
  2. Discussions and implications: I would recommend you highlight how your findings help enrich our understanding. The current version of the discussion section does not sufficiently highlight the contributions of the study
  3. The practical implications are not actionable and speculative at best.
  4. Highly simplistic and obvious your limitations, further research.

Author Response

Dear Reviewer

Thank you for giving us the opportunity to submit a revised draft of the manuscript " The Effect of Value Innovation in the Superior Performance and Sustainable Growth of Telecommunications Sector: Mediation Effect of Customer Satisfaction and Loyalty " for publication in the Sustainability. We appreciate the time and effort you dedicated to providing feedback on our manuscript and are grateful for the insightful comments and valuable improvements to our paper. We have highlighted the changes within the manuscript. Here is a point-by-point response to the reviewers' comments and concerns.

Thank you, and we highly appreciate your kind consideration and cooperation in approving and listing our manuscript for publication shortly. Please see the attachment for details of the revised manuscript.

Comments and Suggestions for Authors

  1. Without a strong engagement in recent studies, the study seems generic and cursory, the arguments seem patched and do not extend much beyond existing literature, and the paper does not seem to provide the type of deep, novel theoretical contributions we typically seek at novel research. Prior literature reviewed in your study looks outdated. You need to strengthen the section with literature/references based on the articles published recently, to convince that your paper examines prior work and has some new/novel paradigm.

Response:

Thank you for the valuable comments. The revised version of the manuscript has added recent citations. Literature review updated and recent researches discussed and cited.

  1. There must be some reasons or explanations for why testing this model is important in Yemen. The linkage between the arguments is either missed or weak However, why do you have to test the impact of value innovation on firm performance and long-term growth by examining the mediation effect 24 of customer satisfaction and loyalty.? What kind of Companies are you talking about? Is it global companies and what is the context? There is no explanation about the industry of in the introduction. I think this should be revised.

Response

Thank you for highlighting these points. The introduction has been updated in the revised version of the manuscript to address the reviewer’s comments. Please refer to the introduction section.  

  1. The introduction contains much general information frequently in more than one section and there is no preamble to the research and its importance what do you want to do.  A further effort is needed in the introductory section, for instance, there is a need to give a general overview of your study knowing fully well that your audience is expanded; also, the latter part of the introduction should have the statement of the problem, research question, and objective, in order.
    - The research question is not clear. Make strong discussions about why the research is important, what motivations are required to complete the research and what gaps you want to search? -I could not understand what are the gaps or contributions of this research. In gap and contribution part makes the case more strongly by depending on the recent literature review about the topic and what is the needed gaps. What was the key contribution(s) of this study that would not have been known from existing research? The arguments should go beyond "few studies investigated these effects, and therefore we wanted to investigate it". For example, "Only a limited number of research papers” " no large-scale study investigates the nature of value innovation in the telecommunications industry.". “and long-term competitive advantage is not investigated yet”. Given These arguments are not sufficient. Even without the evidence from this research, the relationships among study constructs were straightforward and intuitive. The authors should make salient and strengthen this study's unique contributions in the introduction and discussion sections.

Response

Thank you for highlighting these points. The introduction has been updated in the revised version of the manuscript to address the reviewer’s comments. We hope it is clearer. Please refer to the introduction section. 

  1. Literature Review-This is bulky with some irrelevant information; it should be summarized under two to three pages. A well-summarized review of related literature would open the glaring gap and give relevance to the intending study. Make the case more strongly in the literature review as to why you're doing this research. It could be much stronger as you can point to truthfulness as an important construct. The paper could use that. You need to strengthen the section with literature/references based on the articles published recently, to convince that your paper.

Response

we have updated the related work section to address the reviewer’s comment. We believe it is clearer.

  1. you need to enlarge the literature background and references. Some authors are cited many times in the document and support several affirmations and hypotheses with the same references all over again. Please update references with the latest developments in the sector and include more varied sources of knowledge. I am including some references that might help, this does not mean you necessary include them all in the manuscript:
    1. Ibrahim, B. 2021. “Social Media Marketing Activities and Brand Loyalty: A Meta-Analysis Examination.” Journal of Promotion Management. 1–31. doi:10.1080/10496491.2021.1955080
    2. Ibrahim, B., & Aljarah, A. (2021). The era of Instagram expansion: matching social media marketing activities and brand loyalty through customer relationship quality. Journal of Marketing Communications, 1-25.
    3. Ting, H., K. Lim Tan, L. Xin Jean, and C. Jun Hwa. 2020. “What Determines Customers’ Loyalty Towards Telecommunication Service Mediating Roles of Satisfaction and Trust Responsible Tourism View Project Marketing View Project.” Article in International Journal of Services Economics and Management. doi:10.1504/IJSEM.2020.111179.Jacoby, J., and R. W. Chestnut. 1978. Brand Loyalty: Measurement and Management. New York, NY: Wiley.
    4. Chaudhuri, A., and M. Holbrook. 2001. “The Chain of Effects from Brand Trust and Brand Affect to Brand Performance: The Role of Brand Loyalty.” Journal of Marketing 65 (2): 81–93. doi:10.1509/ jmkg.65.2.81.18255

Response

Thank you for the valuable comments. The recommended manuscripts have been added to the revised version of the manuscript.

  1. The theoretical framework also needs strengthening. the innovation of the paper is not enough. For example, what did we learn about Value innovation and The Resources-Based View (RBV)? It is unclear how the theoretical framework really influenced this study. it isn’t used to rationalize any Hs until H7.

Response

We clarified the main aim of the proposed model in the revised version of the manuscript.

  1. Each hypothesis from H1 to H7 should discuss separately with strong arguments

Response

We agree with the reviewer's comment that discussing each hypothesis separately will improve the manuscript. Therefore, we discussed each hypothesis separately.

  1. The screening procedure is explained somewhat convolutely. Can the authors walk the readers through the procedure? This is relevant not only to understand how the study ended up with a certain sample size but also to appreciate the selection that the sample went through. From this section, it's also unclear whether all respondents are reflecting upon the same brand pages, different brand pages, etc.

Response

Thank you for your valuable suggestions. The methodology section has been updated with the screening procedure and the distributed questioners’ and how we get the final sample size.

Reviewer 4 Report

The Authors create a conceptual model to examine the impact of innovation on firm performance by the mediation effect between customer loyalty and customer satisfaction. Partial Least squares structural equation modelling is used to test hypothesis relationships.

The idea of the paper is clear, and it is well-written. However, there are some major and minor details to take into account. I hope you could find these comments helpful.

Major comments

  1. Please check the name of the hypothesis throughout the whole test. H1a, H1b, H1c don’t appear in the text, and there could be more errors. Please check lines 280-295. I would put all this information in another paragraph and introduce it a little bit more. This information is critical to understanding the results of the model.
  2. A deeper linkage between the results and the discussion would be appreciated. Sometimes, it is missed.
  3. The threshold for HTMT assessment results is 0.9. However, Sarstedt et al (2017) mention 0.85 as threshold, and do bootstrapping procedure to assess the discriminant validity. I would suggest bootstrapping procedure as the HTMT values for customer loyalty are quite high.

Sarstedt, M., Ringle, C. M., & Hair, J. F. (2017). Partial least squares structural equation modeling. Handbook of market research, 26(1), 1-40.

Minor details

  1. Have you considered using “PLS-SEM” instead of “PLS-PM” for “partial least squares structural equation modeling”. I would use it also as keyword.
  2. I would appreciate more information about the questionnaire. How many questions does it have? How many questions were used for each hypothesis?
  3. Could you explain a little bit more point 5.4. In table 4, there is no H1b and H1c.
  4. Check the format of the text in lines 61, 62, 63

Author Response

Dear Reviewer

Thank you for giving us the opportunity to submit a revised draft of the manuscript " The Effect of Value Innovation in the Superior Performance and Sustainable Growth of Telecommunications Sector: Mediation Effect of Customer Satisfaction and Loyalty " for publication in the Sustainability. We appreciate the time and effort you dedicated to providing feedback on our manuscript and are grateful for the insightful comments and valuable improvements to our paper. We have highlighted the changes within the manuscript. Here is a point-by-point response to the reviewers' comments and concerns.

Thank you, and we highly appreciate your kind consideration and cooperation in approving and listing our manuscript for publication shortly. Please see the attachment for details of the revised manuscript.

Comments and Suggestions for Authors

The Authors create a conceptual model to examine the impact of innovation on firm performance by the mediation effect between customer loyalty and customer satisfaction. Partial Least squares structural equation modelling is used to test hypothesis relationships. The idea of the paper is clear, and it is well-written. However, there are some major and minor details to take into account. I hope you could find these comments helpful.

Major comments

Please check the name of the hypothesis throughout the whole test. H1a, H1b, H1c don’t appear in the text, and there could be more errors. Please check lines 280-295. I would put all this information in another paragraph and introduce it a little bit more. This information is critical to understanding the results of the model.

Response:

      Thank you for highlighting this typo mistake. The paper reviewed and corrected the hypothesis names to be synchronized with internal review editing.

A deeper linkage between the results and the discussion would be appreciated. Sometimes, it is missed.

Response:

The threshold for HTMT assessment results is 0.9. However, Sarstedt et al (2017) mention 0.85 as threshold, and do bootstrapping procedure to assess the discriminant validity. I would suggest bootstrapping procedure as the HTMT values for customer loyalty are quite high.

Sarstedt, M., Ringle, C. M., & Hair, J. F. (2017). Partial least squares structural equation modeling. Handbook of market research, 26(1), 1-40.

Response:

You have raised an important point here. However, regarding your concerns about the HTMT values, we can clarify that according to Henseler, Ringle, and Sarstedt (2015); Franke and Sarstedt (2019), in interpreting the HTMT threshold values, 0.90 can be interpreted as an upper boundary of acceptable construct correlations, and this condition does not represent a definite violation of discriminant validity. Accordingly, the proposed model has a satisfying the discriminant validity criterion. Please refer to the following references:

Franke, G., & Sarstedt, M. (2019). Heuristics versus statistics in discriminant validity testing: a comparison of four procedures. Internet Research, 29(3), 430-447. doi:10.1108/IntR-12-2017-0515.

Henseler, J., Ringle, C. M., & Sarstedt, M. (2015). A new criterion for assessing discriminant validity in variance-based structural equation modeling. Journal of the Academy of Marketing science, 43(1), 115-135. doi:10.1007/s11747-014-0403-8

Minor details

Have you considered using “PLS-SEM” instead of “PLS-PM” for “partial least squares structural equation modeling”. I would use it also as keyword. I would appreciate more information about the questionnaire. How many questions does it have? How many questions were used for each hypothesis? Could you explain a little bit more point 5.4? In table 4, there is no H1b and H1c.

Response:

Thank you for highlighting these points; we did not notice this problem in the previous version, thanks to you for the reminder, now we have revised the hypothesis section the typo mistakes have been corrected in the revised version of the manuscript. H1b & H1c have been corrected to H6 & H7 as per internal review editing. Furthermore, we have added the adapted constructs, items, and sources along with the instrument design and validation; please refer to section 4 [lines 308 – 344] and Appendix 1 in the revised version of the manuscript.

Check the format of the text in lines 61, 62, and 63.

Response:

Thank you for highlighting this. The format has been checked and updated in the revised version of the manuscript.

Round 2

Reviewer 2 Report

The authors responded correctly to the remarks from the first review and supplemented the text. However, this does not apply to the remark under Ad 4, which reads:
"As stated in the paper (311 to 315)," researcher employed a quantitative method to collect data from Yemeni mobile service providers' employees via a paper-based survey. Data was collected sequentially among telecommunications companies and within each telecommunications company department to avoid duplication and missed collection. "From the description of the methodology, it is clear that the instrument is based on the perception of the respondents, and not on the measurement of some process variables or financial indicators. This approach has a sense in some indicators, such as respondents' frustration level (333). However, research into the impact of value innovation on some of the exact indicators listed in the hypotheses (288 to 295) would be better based on insight into actual data than on respondents' perceptions. to H2, the impact of Value innovation on companies 'sustainable growth (288). Company growth data is more obvious and accurate in financial statements than in respondents' perceptions. The same is true with the H3 which deals with customer loyalty. exactly measured as retention rate, drop-out rate, etc. It is exactly the same with H4 companies' performance, which is actually a complex indicator, composed of several elementary indicators in the field of finance, sales, human resources, etc. Therefore, the best validation of constructs and instruments used in this research is in relation to objective reality, ie data that can be accurately measured. Such validation also provides an answer as to whether the respondents' perception of a phenomenon is consistent with objective reality, which is not always the case for a variety of reasons. The authors need to supplement the research in this regard. "

The author's answer is:
"Response: We agree with the reviewer's comment that the instrument is based on the respondents' perceptions. As we mentioned in the revised version of the manuscript, all measurement items were adapted with slight adjustments from well-known sources of the extant studies in the innovation value. In addition, the proposed model was verified based on the real responses of companies' employees and managers who are more familiar with their companies. Therefore, the measurement items have been added to the revised version of the manuscript. "

This requires some explanation. The remark refers to the application of an instrument based on perception, to the "measurement" of phenomena that can be measured and expressed exactly. For example, customer loyalty can be expressed using certain indicators, such as drop-out rate (number of customers who stopped using the service after certain time periods, before or after the expiration of the contract, divided by the total number of customers), retention index (number of customers who renewed the contract, divided by the total number of customers) etc. It is similar with financial indicators. In such cases, the instrument should be validated in the exact determination of the values ​​of the phenomena they determine and based on the perception of the respondents, regardless of the fact that the respondents are experts in the problem domain, as stated by the authors. It is also a method of checking the accuracy of perception.
Clearly, there are phenomena that can only be assessed by perception, such as frustration in communication, stress, subjective user experience and the like.
If this view is acceptable to the authors, I suggest supplementing the limitation of research in paper.

Author Response

Dear Reviewer 2

We appreciate the time and effort you dedicated to providing feedback on our manuscript and are grateful for the insightful comments and valuable improvements to our paper. We have now carefully revised our manuscript according to the comments as indicated below.

Comments and Suggestions for Authors

The authors responded correctly to the remarks from the first review and supplemented the text. However, this does not apply to the remark under Ad 4, which reads:
"As stated in the paper (311 to 315)," researcher employed a quantitative method to collect data from Yemeni mobile service providers' employees via a paper-based survey. Data was collected sequentially among telecommunications companies and within each telecommunications company department to avoid duplication and missed collection. "From the description of the methodology, it is clear that the instrument is based on the perception of the respondents, and not on the measurement of some process variables or financial indicators. This approach has a sense in some indicators, such as respondents' frustration level (333). However, research into the impact of value innovation on some of the exact indicators listed in the hypotheses (288 to 295) would be better based on insight into actual data than on respondents' perceptions. to H2, the impact of Value innovation on companies 'sustainable growth (288). Company growth data is more obvious and accurate in financial statements than in respondents' perceptions. The same is true with the H3 which deals with customer loyalty. exactly measured as retention rate, drop-out rate, etc. It is exactly the same with H4 companies' performance, which is actually a complex indicator, composed of several elementary indicators in the field of finance, sales, human resources, etc. Therefore, the best validation of constructs and instruments used in this research is in relation to objective reality, ie data that can be accurately measured. Such validation also provides an answer as to whether the respondents' perception of a phenomenon is consistent with objective reality, which is not always the case for a variety of reasons. The authors need to supplement the research in this regard. "

The author's answer is:
"Response: We agree with the reviewer's comment that the instrument is based on the respondents' perceptions. As we mentioned in the revised version of the manuscript, all measurement items were adapted with slight adjustments from well-known sources of the extant studies in the innovation value. In addition, the proposed model was verified based on the real responses of companies' employees and managers who are more familiar with their companies. Therefore, the measurement items have been added to the revised version of the manuscript. "

This requires some explanation. The remark refers to the application of an instrument based on perception, to the "measurement" of phenomena that can be measured and expressed exactly. For example, customer loyalty can be expressed using certain indicators, such as drop-out rate (number of customers who stopped using the service after certain time periods, before or after the expiration of the contract, divided by the total number of customers), retention index (number of customers who renewed the contract, divided by the total number of customers) etc. It is similar with financial indicators. In such cases, the instrument should be validated in the exact determination of the values ​​of the phenomena they determine and based on the perception of the respondents, regardless of the fact that the respondents are experts in the problem domain, as stated by the authors. It is also a method of checking the accuracy of perception. Clearly, there are phenomena that can only be assessed by perception, such as frustration in communication, stress, subjective user experience and the like. If this view is acceptable to the authors, I suggest supplementing the limitation of research in paper.

Authors response:

We agree that this is a potential limitation of the study. We have added this as a limitation on the revised version of the manuscript. Please refer to lines [6222908].

Reviewer 3 Report

1-“The introduction section could be further strengthened. For instance, why do we need to know about how value innovation affects firm performance and long-term?

Why is it important to investigate the underlying mechanisms of how to value innovation promotes firm performance and long-term through customer satisfaction and loyalty?”

2- “Why do author(s) focus on satisfaction and loyalty as mediator among others?”

3- “The author(s) need to conduct a non-response bias test. Also, they need to test common method bias test.”

4- “Mean value and SD should be reported.”

5- “The authors should discuss how the findings contribute to the theories used in this research (about Value innovation and The Resources-Based View (RBV)).”

6- Findings: The authors should combine the findings and arguments from previous literature appropriately.  The objective is to develop a clear and convincing text, in which the hypotheses they formulate are the logical conclusions of the storyline of the paper. I cannot capture any further key information regarding variables and measures

Author Response

Dear Reviewer 3

We appreciate the time and effort you dedicated to providing feedback on our manuscript and are grateful for the insightful comments and valuable improvements to our paper. We have now carefully revised our manuscript according to the comments as indicated below.

Comments and Suggestions for Authors

  • “The introduction section could be further strengthened. For instance, why do we need to know about how value innovation affects firm performance and the long term? Why is it important to investigate the underlying mechanisms of how to value innovation promotes firm performance and long-term through customer satisfaction and loyalty?”

Authors response:

Thank you for your comment. The introduction section has been updated to address your comment. Please refer to lines [119-140], in the revised version of the manuscript.

2- “Why do author(s) focus on satisfaction and loyalty as mediators among others?”

Authors response:

The justification for using the satisfaction and loyalty variables as mediators in this study is provided in section 3 “Research Model and Hypotheses Development”. Please refer to lines [289 – 336].

3- “The author(s) need to conduct a non-response bias test. Also, they need to test common method bias test.”

Authors response:

Thank you for pointing this out. The required tests are provided. Please refer to lines [424 to 428], and lines [440-446].

4- “Mean value and SD should be reported.”

Authors response:

While we appreciate the reviewer’s feedback, we respectfully disagree. There is no descriptive analysis included in the manuscript therefore, the mean and SD tests are not applicable in the context of this study.

5- “The authors should discuss how the findings contribute to the theories used in this research (about Value innovation and The Resources-Based View (RBV)).”

Authors response:

Thank you for your valuable comment. The discussion section has been enriched with more cited research to compare the findings of this study with the previous studies. Please refer to lines [515-525], in the revised version of the manuscript.

6- Findings: The authors should combine the findings and arguments from previous literature appropriately.  The objective is to develop a clear and convincing text, in which the hypotheses they formulate are the logical conclusions of the storyline of the paper. I cannot capture any further key information regarding variables and measures.

Authors response:

Thank you for your valuable comment. The discussion section has been enriched with more cited research to compare the findings of this study with the previous studies. Please refer to lines [622-629], in the revised version of the manuscript.
